# PROGRAMMING WITH A DIFFERENTIABLE FORTH INTERPRETER

**Matko Bošnjak, Tim Rocktäschel, Jason Naradowsky & Sebastian Riedel**
Department of Computer Science
University College London
London, UK
{m.bosnjak, t.rocktaschel, j.narad, s.riedel}@cs.ucl.ac.uk

## ABSTRACT

There are families of neural networks that can learn to compute any function, provided sufficient training data. However, given that in practice training data is scarce for all but a small set of problems, a core question is how to incorporate prior knowledge into a model. Here we consider the case of prior *procedural* knowledge, such as knowing the overall recursive structure of a sequence transduction program or the fact that a program will likely use arithmetic operations on real numbers to solve a task. To this end we present a differentiable interpreter for the programming language Forth. Through a neural implementation of the dual stack machine that underlies Forth, programmers can write program sketches with slots that can be filled with behaviour trained from program input-output data. As the program interpreter is end-to-end differentiable, we can optimize this behaviour directly through gradient descent techniques on user specified objectives, and also integrate the program into any larger neural computation graph. We show empirically that our interpreter is able to effectively leverage different levels of prior program structure and learn complex transduction tasks such as sequence sorting or addition with substantially less data and better generalisation over problem sizes. In addition, we introduce neural program optimisations based on symbolic computation and parallel branching that lead to significant speed improvements.

## 1 INTRODUCTION

A central goal of Artificial Intelligence is the creation of machines that learn as effectively from human instruction as they do from data. A recent and important step towards this goal is the invention of neural architectures that can learn to perform algorithms akin to traditional computers, using primitives such as memory access and stack manipulation (Graves et al., 2014; Joulin & Mikolov, 2015; Grefenstette et al., 2015; Kaiser & Sutskever, 2015; Kurach et al., 2015; Graves et al., 2016). These architectures can be trained through standard gradient descent methods, and enable machines to learn complex behavior from input-output pairs or program traces. In this context the role of the human programmer is often limited to providing training data. However, for many tasks training data is scarce. In these cases the programmer may have *partial* procedural background knowledge: one may know the rough structure of the program, or how to implement several sub-routines that are likely necessary to solve the task. For example, in visual programming, a user often knows a rough sketch of what they want to do, but need to fill in the specific components. In programming by demonstration (Lau et al., 2001) and programming with query languages (Neelakantan et al., 2015a) a user conforms to a larger set of conditions on the data, and needs to settle details. In all these scenarios, the question then becomes how to exploit this type of prior knowledge when learning algorithms.

To address the above question we present an approach that enables programmers to inject their procedural background knowledge into a neural network. In this approach the programmer specifies a program *sketch* (Solar-Lezama et al., 2005) in a traditional programming language. This sketch defines one part of the neural network behaviour. The other part is learned using training data. The core insight that enables this approach is the fact that most programming languages can be formulated

in terms of an abstract machine that executes the commands of the language. We implement these machines as neural networks, constraining parts of the networks to follow the sketched behaviour. The resulting neural programs are consistent with our prior knowledge and optimised with respect to the training data.

In this paper we focus on the programming language Forth (Brodie, 1980), a simple yet powerful stack-based language that is relatively close to machine code but enables modular programs and facilitates abstraction. Underlying Forth's semantics is a simple abstract machine. We introduce the Forth Neural Abstract Machine ($\partial 4$), an implementation of this machine that is differentiable with respect to the transition it executes at each time step, as well as distributed input representations in the machine buffers. As sketches that users define are also differentiable, any underspecified program content contained within the sketch can be trained through backpropagation.

For two neural programming tasks introduced in previous work (Reed & de Freitas, 2015) we present Forth sketches that capture different degrees of prior knowledge. For example, we define only the general recursive structure of a sorting problem. We show that given only input-output pairs, $\partial 4$ can learn to fill the sketch and generalise well to problems of unseen size. We also use $\partial 4$ to investigate the type and degree of structure necessary when solving tasks, and show how symbolic execution can significantly improve execution time when applicable.

The contribution of our work is fourfold: i) we present a neural implementation of a dual stack machine underlying Forth, ii) we introduce Forth sketches for programming with partial procedural background knowledge, iii) we apply Forth sketches as a procedural prior on learning algorithms from data, and iv) we introduce program code optimisations based on symbolic execution that can speed up neural execution.

## 2 THE FORTH ABSTRACT MACHINE

Forth is a simple Turing-complete stack-based programming language (ANSI, 1994; Brodie, 1980). Its underlying abstract machine is represented by a state $S = (D, R, H, c)$, which contains two stacks: a data evaluation pushdown stack (*data stack*) $D$ holds values for manipulation, and a return address pushdown stack (*return stack*) $R$ assists with return pointers and subroutine calls. These are accompanied by a *heap* or random memory access buffer $H$, and a program counter $c$.

A Forth program $P$ is a flat sequence of Forth *words* (i.e. commands) $P = w_1 \ldots w_n$. The role of a word varies, encompassing language keywords, primitives, and user-defined subroutines (e.g. DROP, to discard the top element of the stack, or DUP, to duplicate the top element of the stack).[1] Each word $w_i$ defines a transition function between machine states, $w_i : S \rightarrow S$. Therefore, a program $P$ itself defines a transition function by simply applying the word at the current program counter to the current state. Although usually considered as a part of the heap $H$, we consider Forth programs $P$ separately to ease the analysis.

An example of a Forth program that implements the Bubble sort algorithm, is shown in Listing 1, and a detailed description of how this program is executed by the Forth abstract machine is provided in Appendix B. Notice that while Forth provides common control structures such as looping and branching, these can always be reduced to low-level code that uses jumps and conditional jumps (using the words BRANCH and BRANCH0, respectively). Likewise, we can think of sub-routine definitions as code blocks tagged with a label, and their invocation amounts to jumping to the tagged label.

## 3 THE DIFFERENTIABLE FORTH ABSTRACT MACHINE

When a programmer writes a Forth program, they define a sequence of Forth words, *i.e.*, a sequence of *known* state transition functions. In other words, the programmer knows *exactly* how computation should proceed. To accommodate for cases when the developer's procedural background knowledge is incomplete, we extend Forth to support the definition of a program *sketch*. As is the case with Forth programs, sketches are sequences of transition functions. However, a sketch may contain transition functions whose behavior is learned from data.

---

[1] In this work, we restrict ourselves to a subset of all Forth words, detailed in Appendix A.

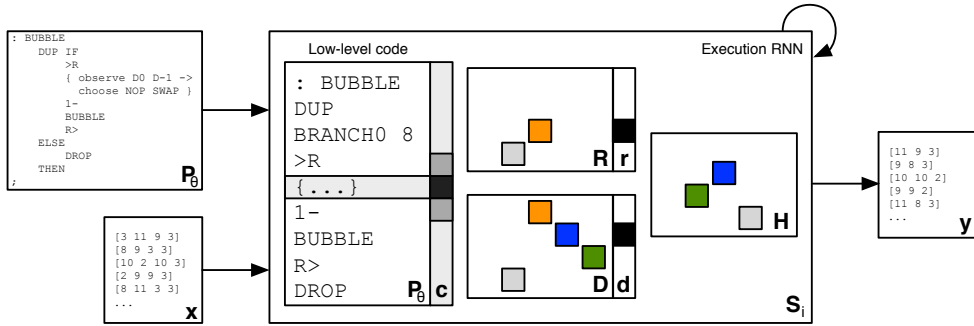

Figure 1: Neural Forth Abstract Machine. Forth sketch $\mathbf{P}_\theta$ is translated to a low-level code, with the slot $\{\ldots\}$ substituted by a parametrised neural network. The slot is learnt from input-output examples $(\mathbf{x}, \mathbf{y})$ through the differentiable machine whose state $\mathbf{S}_i$ comprises the low-level code and program counter $\mathbf{c}$, data stack $\mathbf{D}$ (with pointer $\mathbf{d}$), return stack $\mathbf{R}$ (with pointer $\mathbf{r}$), and the heap $\mathbf{H}$.

To learn the behaviour of transition functions within a program we would like the machine output to be differentiable with respect to these functions (and possibly representations of inputs to the program). This enables us to choose parameterized transition functions such as neural networks, and efficiently train their parameters through backpropagation and gradient methods. To this end we first provide a continuous representation of the state of a Forth abstract machine. We then present a recurrent neural network (RNN) that models program execution on this machine, parametrised by the transition functions at each time step. Lastly, we discuss optimizations based on symbolic execution and the interpolation of conditional branches.

## 3.1 MACHINE STATE ENCODING

We map the symbolic machine state $S = (D, R, H, c)$ to a continuous representation $\mathbf{S} = (\mathcal{D}, \mathcal{R}, \mathbf{H}, \mathbf{c})$ – into two differentiable stacks (with pointers), the data stack $\mathcal{D} = (\mathbf{D}, \mathbf{d})$ and the return stack $\mathcal{R} = (\mathbf{R}, \mathbf{r})$, a heap $\mathbf{H}$, and an attention vector $\mathbf{c}$ indicating which word of the sketch $\mathbf{P}_\theta$ is being executed at the current time step. All three memory structures, the data stack, the return stack and the heap, are based on differentiable flat memory buffers $\mathbf{M} \in \{\mathbf{D}, \mathbf{R}, \mathbf{H}\}$, where $\mathbf{D}, \mathbf{R}, \mathbf{H} \in \mathbb{R}^{l \times v}$, for a stack size $l$ and a value size $v$. Each has a well-defined, differentiable read operation:

$$\text{read}_\mathbf{M}(\mathbf{a}) = \mathbf{a}^T \mathbf{M} \tag{1}$$

and write operation:

$$\text{write}_\mathbf{M}(\mathbf{x}, \mathbf{a}) : \mathbf{M} \leftarrow \mathbf{M} - (\mathbf{a} \otimes \mathbf{1}) \odot \mathbf{M} + \mathbf{x} \otimes \mathbf{a} \tag{2}$$

akin to the Neural Turing Machine (NTM) memory (Graves et al., 2014), where $\otimes$ is the outer product, $\odot$ is the Hadamard product, and $\mathbf{a}$ is the address pointer.[2] In addition to the memory buffers $\mathbf{D}$ and $\mathbf{R}$, the data stack and the return stack contain pointers to the current top-of-the-stack (TOS) element $\mathbf{d}, \mathbf{r} \in \mathbb{R}^l$. This allows us to implement pushing as writing a value $\mathbf{x}$ into $\mathbf{M}$ and incrementing the TOS pointer as follows:

$$\text{push}_\mathbf{M}(\mathbf{x}) : \text{write}_\mathbf{M}(\mathbf{x}, \mathbf{p}) \qquad [\text{side-effect:} \mathbf{p} \leftarrow \text{inc}(\mathbf{p})] \tag{3}$$

where $\mathbf{p} \in \{\mathbf{d}, \mathbf{r}\}$, $\text{inc}(\mathbf{p}) = \mathbf{p}^T \mathbf{R}^+$, and $\text{dec}(\mathbf{p}) = \mathbf{p}^T \mathbf{R}^-$. and $\mathbf{R}^+$ and $\mathbf{R}^-$ are increment and decrement matrices (left and right circular shift matrices).

Likewise, popping is realized by multiplying the TOS pointer and the memory buffer, and decreasing the TOS pointer:

$$\text{pop}_\mathbf{M}(\,) = \text{read}_\mathbf{M}(\mathbf{p}) \quad [\text{side-effect:} \mathbf{p} \leftarrow \text{dec}(\mathbf{p})] \tag{4}$$

Finally, the program counter $\mathbf{c} \in \mathbb{R}^p$ is a vector that, when one-hot, points to a single word in a program of length $p$, and is equivalent to the $c$ vector of the symbolic state machine.[3] We will use $\mathcal{S}$ to denote the space of all continuous representations $\mathbf{S}$.

---

[2] The equal widths of $\mathbf{H}$ and $\mathbf{D}$ allow us to directly move vector representations of values between the heap and the stack.

[3] During training $\mathbf{c}$ can become distributed, and is considered as attention over the program code.

```
0 : BUBBLE ( a1 ... an n-1 -- one pass )
1     DUP IF >R
2         OVER OVER < IF SWAP THEN
3         R> SWAP >R 1- BUBBLE R>
4     ELSE
5         DROP
6     THEN
7 ;
8 : SORT ( a1 .. an n -- sorted  )
9   1- DUP 0 DO >R R@ BUBBLE R> LOOP DROP
10 ;
11 2 4 2 7 4 SORT \ Example call
```

Listing 1: BubbleSort in Forth.

```
0 : BUBBLE ( a1 ... an n-1 -- one pass )
1     DUP IF >R
2         { observe D0 D-1 -> permute D-1 D0 R0 }
3         1- BUBBLE R>
4         \ ** Alternative sketch **
5         \ { observe D0 D-1 -> choose NOP SWAP }
6         \ R> SWAP >R 1- BUBBLE R>
7     ELSE
8         DROP
9     THEN
10 ;
```

Listing 2: BUBBLE sketch with trainable permutation (trainable comparison in comments).

**Neural Forth Words**  It is straightforward to convert Forth words, defined as functions on discrete machine states, to functions operating on the continuous space $\mathcal{S}$. For example, consider the Forth word DUP, which duplicates the TOS element of the data stack. A differentiable version works by first calculating the value $\mathbf{e}$ on the TOS address of $\mathbf{D}$, as $\mathbf{e} = \mathbf{d}^T \mathbf{D}$. It then shifts the stack pointer via $\mathbf{d} \leftarrow \text{inc}(\mathbf{d})$, and writes $\mathbf{e}$ to $\mathbf{D}$ using $\text{write}_{\mathbf{D}}(\mathbf{e}, \mathbf{d})$. We present the complete description of implemented Forth Words in Appendix A and their differentiable counterparts in Appendix C.

## 3.2  FORTH SKETCHES

We define a Forth sketch $\mathbf{P}_\theta$ as a sequence of continuous transition functions $\mathbf{P} = \mathbf{w}_1 \ldots \mathbf{w}_n$. Here, $\mathbf{w}_i \in \mathcal{S} \to \mathcal{S}$ either corresponds to a neural Forth word, or is a trainable transition function. We will call these trainable functions *slots*, as they correspond to underspecified "slots" in the program code that need to be filled by learned behaviour.

We allow users to define a slot $\mathbf{w}$ by specifying a pair of a state encoder $\mathbf{w}_{\text{enc}}$ that produces a latent representation $\mathbf{h}$ of the current machine state using a multi-layer perceptron, and a decoder $\mathbf{w}_{\text{dec}}$ that consumes this representation to produce the next machine state. We hence have $\mathbf{w} = \mathbf{w}_{\text{dec}} \circ \mathbf{w}_{\text{enc}}$. To use slots within Forth program code we introduce a notation that reflects this decomposition. In particular, slots are defined using the syntax { encoder -> decoder } where encoder and decoder are specifications of the corresponding slot parts as described below.

**Encoders**  We provide the following options for encoders:

- static: produces a static representation, independent of the actual machine state.
- observe $e_1 \ldots e_m$: concatenates the elements $e_1 \ldots e_m$ of the machine state. An element can be a stack item D$i$ at relative index $i$, a return stack item R$i$, etc.

**Decoders**  Users can specify the following decoders:

- choose $w_1 \ldots w_m$: chooses from the Forth words $w_1 \ldots w_m$. Takes an input vector $\mathbf{h}$ of length $m$ to produce a weighted combination of machine states $\sum_i^m h_i \mathbf{w_i}(\mathbf{S})$.
- manipulate $e_1 \ldots e_m$: directly manipulates the machine state elements $e_1 \ldots e_m$ by writing the appropriately reshaped output of the encoder over the machine state elements with $\text{write}_{\mathbf{M}}$.
- permute $e_1 \ldots e_m$: permutes the machine state elements $e_1 \ldots e_m$ via a linear combination of $m!$ state vectors.

## 3.3  THE EXECUTION RNN

We model execution using an RNN which produces a state $\mathbf{S}_{i+1}$ conditioned on a previous state $\mathbf{S}_i$. It does so by first passing the current state to each function $\mathbf{w}_i$ in the program, and then weighing each of the produced next states by the component of the program counter vector $\mathbf{c}_i$ that corresponds to program index $i$, effectively using $\mathbf{c}$ as an attention vector over code. Formally we have:

$$\mathbf{S}_{i+1} = \text{RNN}(\mathbf{S}_i, \mathbf{P}_\theta) = \sum_i \mathbf{c}_i \mathbf{w}_i(\mathbf{S}_i) \tag{5}$$

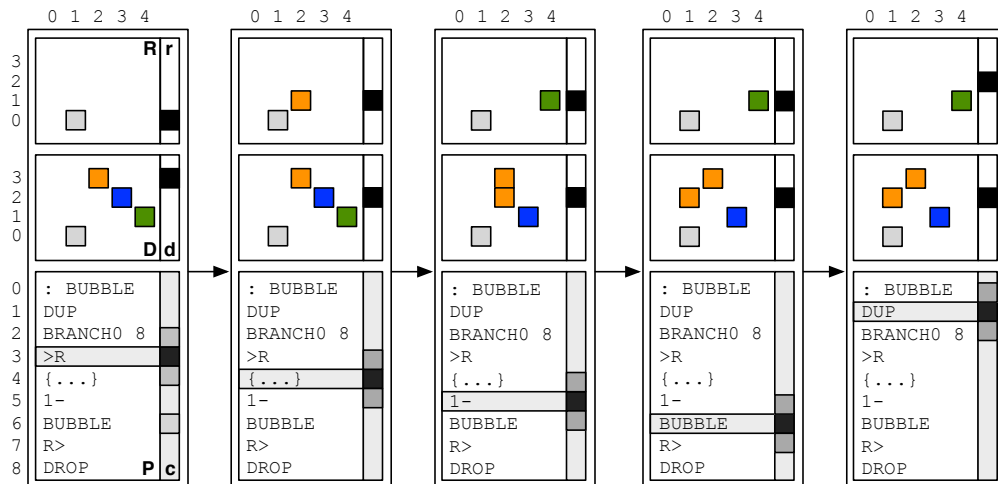

Figure 2: $\partial 4$ Segment of the RNN execution of a Forth sketch in Listing 2. The pointers ($\mathbf{d}$, $\mathbf{r}$) and values (rows of $\mathbf{R}$ and $\mathbf{D}$) are all in one-hot state (colors simply denote values observed, defined by the top scale), while the program counter maintains the uncertainty. Subsequent states are discretised for clarity. Here the slot $\{\ldots\}$ has learned its optimal behaviour.

Clearly this recursion, and its final state, are differentiable with respect to the program code $\mathbf{P}$, and its inputs. Furthermore, for differentiable Forth programs it is easy to show that the final state of this RNN will correspond to the final state of a symbolic execution.

### 3.3.1  $\partial 4$ EXECUTION OF A BUBBLESORT SKETCH

Listing 2 defines the BUBBLE word as a sketch capturing several types of prior knowledge. In this section we describe the PERMUTE sketch. In it, we assume BUBBLE involves a recursive call, that terminates at length 1, and that the next BUBBLE call takes as input some function of the current length and the top two stack elements.

The input to this sketch are the sequence to be sorted and its length decremented by one, $n-1$ (line 0). These inputs are expected on the data stack. After the length ($n-1$) is duplicated for further use with DUP, the machine tests whether it is non-zero (using IF, which consumes the TOS during the check). If $n-1 > 0$, it is stored on the $R$ stack for future use (line 1).

At this point (line 2) the programmer only knows that a decision must be made based on the top two data stack elements D0 and D-1 (comparison elements), and the top return stack, R0 (length decremented by 1). Here the precise nature of this decision is unknown, but is limited to variants of permutation of these elements, the output of which produce the input state to the decrement -1 and the recursive BUBBLE call (line 3). At the culmination of the call, R0, the output of the learned slot behavior, is moved onto the data stack using R>, and execution proceeds to the next step.

Figure 2 illustrates how portions of this sketch are executed on the $\partial 4$ RNN. The program counter initially resides at >R (line 3 in $\mathbf{P}$), as indicated by the vector $\mathbf{c}$, next to program $\mathbf{P}$. Both data and return stacks are partially filled ($\mathcal{R}$ has 1 element, $\mathcal{D}$ has 4), and we show the content both through horizontal one-hot vectors and their corresponding integer values (color coded). The vectors $\mathbf{d}$ and $\mathbf{r}$ point to the top of both stacks, and are in a one-hot state as well. In this execution trace the slot at line 4 is already showing optimal behaviour: it remembers the element on the return stack (4) is larger, and executes BUBBLE on the remaining sequence with the counter $n$ subtracted by one, to 1.

### 3.4  PROGRAM CODE OPTIMIZATIONS

The $\partial 4$ RNN requires one time step per transition. After each time step the program counter is either incremented or decremented by one, or explicitly set or popped from the stack to jump. In turn a new machine state is calculated by executing all words in the program, and then weighting the result states by the activation of the program counter at the given word. This parallel execution of all words

is expensive, and it is therefore advisable to avoid full RNN steps wherever possible. We use two strategies to significantly speed-up $\partial 4$.

**Symbolic Execution**    Whenever we have a sequence of Forth words that contains no branch entry or exit points, we collapse this sequence to a single transition. We do this using symbolic execution (King, 1976): we first fill the stacks and heap of a standard Forth abstract machine with *symbols* representing arbitrary values (e.g. $D = d_1 \ldots d_l$ and $R = r_1 \ldots r_l$), and execute the sequence of Forth words on the machine. This results in a new symbolic state. We use this state, and its difference to the original state, to derive the transition function of the complete sequence. For example, the sequence R> SWAP >R that swaps the top the data stack with the top of the return stack yields the symbolic state $D = r_1 d_2 \ldots d_l$. and $R = d_1 r_2 \ldots r_l$. Compared to the initial state we have only changed the top elements on both stacks, and hence the neural transition will only need to swap the top elements of $\mathbf{D}$ and $\mathbf{R}$.

**Interpolation of If-Branches**    When symbolic execution hits a branching point we generally cannot simply continue execution, as the branching behaviour will depend on the current machine state and we cannot symbolically resolve it. However, for branches arising from if-clauses that involve no function calls or loop structures, we can still avoid giving control back to the program counter and evaluating all words. We simply execute both branches in parallel, and then let the resulting state be the sum of the output states of both branches, weighted by the score given to the symbol TRUE expected on top of the data stack.

## 3.5    TRAINING

Our training procedure assumes input-output pairs of machine start and end states $(\mathbf{x}_i, \mathbf{y}_i)$ only. The output $\mathbf{y}_i$ defines a target memory $\mathbf{Y}_i^D$ and a target pointer $\mathbf{y}_i^d$ on the data stack $\mathbf{D}$, Additionally, we may have a mask $\mathbf{K}_i$ that indicates which components of the stack should be assessed and which should be ignored. For example, we do not care about values in the stack buffer above the target stack depth, dependent on $\mathbf{y}_i^d$. We use $\mathbf{D}_T(\theta, \mathbf{x}_i)$ and $\mathbf{d}_T(\theta, \mathbf{x}_i)$ to denote the final state of $\mathbf{D}$ and $\mathbf{d}$ after $T$ steps of execution RNN, when using initial state $\mathbf{x}_i$, and define the loss function:

$$\mathcal{L}(\theta) = \sum_i \mathbf{K}_i \odot (\mathbf{D}_T(\theta, \mathbf{x}_i) - \mathbf{Y}_i^D)^2 + \sum_i (\mathbf{d}_T(\theta, \mathbf{x}_i) - \mathbf{y}_i^d)^2 \tag{6}$$

We can use backpropagation and any variant of SGD to optimise our loss function. Note that it is trivial to also provide supervision of the intermediate states (trace-level), as done by the Neural Program Interpreter Reed & de Freitas (2015).

## 4    EXPERIMENTS

We test $\partial 4$ on the sorting and addition tasks presented in Reed & de Freitas (2015) with varying levels of program structure. For each problem we introduce two sketches.

The parameters of each sketch are trained using Adam (Kingma & Ba, 2014), with gradient clipping and gradient noise (Neelakantan et al., 2015b). Hyperparameters were tuned via random search on a development variant of each task, for 1000 epochs, repeating each experiment 5 times. During testing we employ memory element discretisation, replacing differentiable stacks and pointers with their discrete counterparts, and effectively allowing the trained model to generalize to any sequence length if the correct sketch behavior has been learned.

To illustrate the generalization ability of this architecture, we compare against a Seq2Seq (Sutskever et al., 2014) baseline. All Seq2Seq models are single-layer, with a hidden size of 50, trained similarly for 1000 epochs using Adam.

## 4.1    SORTING

Sorting sequences of digits is a hard task for RNNs such as LSTMs, as they fail to generalize to sequences that are marginally longer than the ones they have been trained on (Reed & de Freitas,

| Test len. | | 8 | | | 64 | |
|---|---|---|---|---|---|---|
| Train len. | 2 | 3 | 4 | 2 | 3 | 4 |
| Seq2Seq | 26.2 | 29.2 | 39.1 | 13.3 | 13.6 | 15.9 |
| Permute | 100.0 | 100.0 | 100.0 | 100.0 | 100.0 | 100.0 |
| Compare | 100.0 | 100.0 | - | 100.0 | 100.0 | - |

Table 1: Accuracy of Permute and Compare sketches in comparison to a Seq2Seq baseline on the sorting problem.

2015). We investigate several strong priors based on BubbleSort for this transduction task and present two $\partial 4$ sketches that enable us to learn sorting from only few training examples.

1. PERMUTE. The PERMUTE sketch specifies that three elements (the top two elements of the stack, and the top of the return stack) must be permuted based on the former's values. Both the value comparison and the permutation behavior must be learned from input-output examples. Code for this sketch is shown in Listing 2.

2. COMPARE. The COMPARE sketch provides additional prior procedural knowledge to the model. In contrast to PERMUTE, only the comparison between the top two elements on the stack must be learned. It is shown in the Listing 2 comments (lines 5 and 6),

In both sketches, the outer loop can be specified in $\partial 4$ (Listing 1, line 9), which repeatedly calls a function BUBBLE. In doign so, it defines sufficient structure so that the behavior of the network is invariant to the input sequence length.

**Quantitative Evaluation on BubbleSort** A quantitative comparison of our models on the BubbleSort task is provided in Table 1. For a given test sequence length, we vary the training set sizes to illustrate the model's ability to generalize to sequences longer than those it observed during training. Here $\partial 4$ is shown to quickly learn the correct sketch behavior, and is able to generalize perfectly to sort sequences of 64 elements after observing only two element sequences during training. In comparison, the Seq2Seq baseline falters when attempting similar generalizations, and performs close to chance when tested on the longer sequences. The exception to $\partial 4$'s flawless performance on the BubbleSort task arises from computational difficulties when training from longer sequence lengths (as indicated by the absence of length 4 training set results when using the COMPARE sketch). We discuss this issue further in Section 5.

When measuring the performance of the model as the number of training *instances* varies, we can observe the benefit of additional prior knowledge to the optimization process. We show this in Figure 3a, on both train and test accuracy, using train sequences of length 3 and test sequences of length 8. When prior knowledge is provided (COMPARE), the model quickly maximizes the training accuracy. Providing less structure (PERMUTE) results in lower training accuracy when only a few training examples have been observed. However, with additional training instances both sketches learn the correct behavior and generalize equally well.

**Qualitative Analysis on BubbleSort** It is interesting to analyse the program counter traces, depicted in Figure 4. The trace follows a single example from start, to middle, and the end of the training process. In the beginning of training, the program counter starts to deviate from the one-hot representation in the first 20 steps (not observed in the figure due to unobservable changes), and after 2 iterations of SORT, $\partial 4$ fails to correctly determine the next word. After a few training epochs $\partial 4$ learns better permutations which enables the algorithm to take crisp decisions and halt in the correct state.

**Program Code Optimizations** We measure the runtime of BubbleSort on sequences of varying length with and without the optimizations described in Section 3.4. The results of ten repeated runs are shown in Figure 3b and demonstrate large relative improvements for symbolic execution and interpolation of if-branches compared to non-optimized $\partial 4$ code.

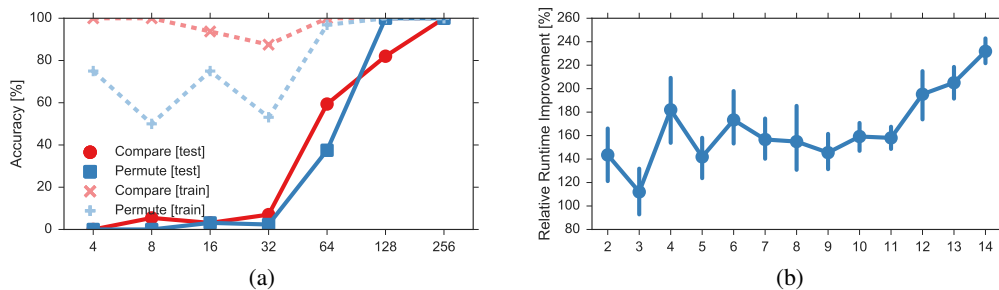

(a)  (b)

Figure 3: Train and test accuracy for varying number of training examples (a) and relative speed improvements of program code optimizations for different input sequence lengths (b).

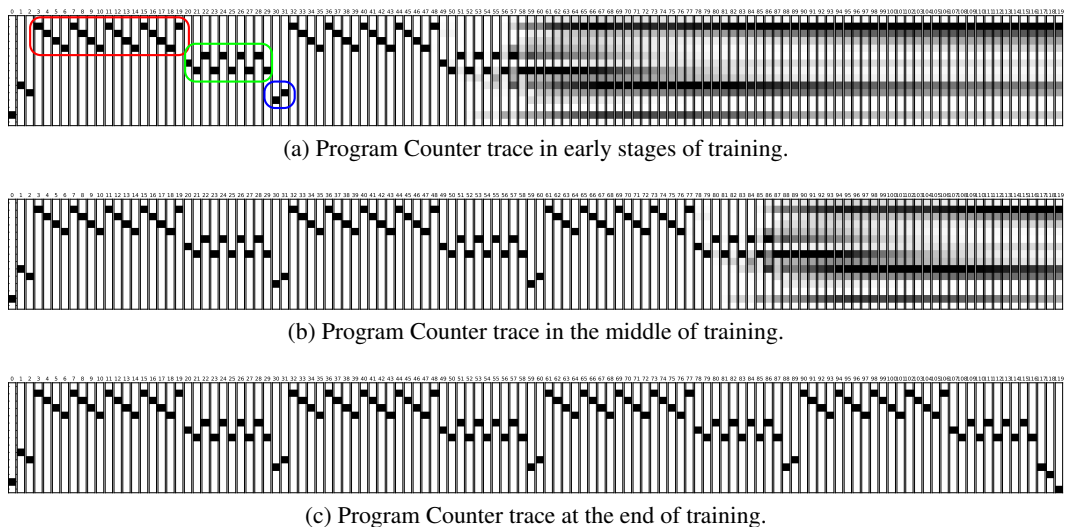

(a) Program Counter trace in early stages of training.

(b) Program Counter trace in the middle of training.

(c) Program Counter trace at the end of training.

Figure 4: Program Counter traces for a single example at different stages of training BubbleSort in Listing 2 (red: successive recursion calls to BUBBLE, green: successive returns from the recursion, and blue: calls to SORT). The last element in the last row is the halting command, which only gets executed after learning the correct slot behavior.

## 4.2   ADDITION

Next we applied $\partial 4$ to the problem of learning to add two numbers of n digits each. We rely on the standard elementary school addition algorithm, where the goal is to iterate over pairs of the aligned digits, calculating the sum of each to yield the sum of the original numbers. The key complication arises when two digits sum to a two-digit number, requiring that the correct extra digit (a *carry*) be carried over to the subsequent column.

Here $\partial 4$ will assume aligned pairs of digits as input, with a carry for the least significant digit (potentially $0$), and the length of the respective numbers. The sketches define the high-level operations through recursion, leaving the core addition to be learned from data.

The specified high-level behavior includes the recursive call template and the halting condition of the recursion (no remaining digits, line 1-2). The underspecified addition operation must take three digits from the previous call, the two digits to sum and a previous carry, and produce a single digit (the sum) and the resultant carry. We introduce two sketches for inducing this behavior:

1.  MANIPULATE. This sketch provides little prior procedural knowledge as it directly manipulates the $\partial 4$ machine state, filling in a carry and the result digits, based on the top three elements on the data stack (two digits and the carry). It is described in Listing 3.

| Test len. | 8 | | | 64 | | |
|---|---|---|---|---|---|---|
| Train len. | 2 | 4 | 8 | 2 | 4 | 8 |
| Seq2Seq | 37.9 | 57.8 | 99.8 | 15.0 | 13.5 | 13.3 |
| Choose | 100.0 | 100.0 | 100.0 | 100.0 | 100.0 | 100.0 |
| Manipulate | 100.0 | – | – | – | – | – |

Table 2: Accuracy of Choose and Manipulate sketches in comparison to a Seq2Seq baseline on the addition problem. Note that lengths corresponds to the length of the input sequence.

```
0 | : ADD-DIGITS
  |   ( a1 b1...an bn carry n -- r1 r2...r_{n+1} )
1 |   DUP 0 = IF
2 |     DROP
3 |   ELSE
4 |     >R \ put n on R
5 |     { observe D0 D-1 D-2 -> manipulate D-1 D-2 }
6 |     DROP R> 1- SWAP >R  \ new_carry n-1
7 |     ADD-DIGITS \ call add-digits on n-1 subseq.
8 |     R> \ put remembered results back on the stack
9 |   THEN
10| ;
```

Listing 3: Manipulate sketch for Elementary Addition. Input data is used to fill data stack externally

```
0 | : ADD-DIGITS
1 |   DUP 0 = IF
2 |     DROP
3 |   ELSE
4 |     >R
5 |     { observe D0 D-1 D-2 -> choose 0 1 }
6 |     { observe D-1 D-2 D-3
  |       -> choose 0 1 2 3 4 5 6 7 8 9 }
7 |     >R SWAP DROP SWAP DROP SWAP DROP R>
8 |     R> 1- SWAP >R
9 |     ADD-DIGITS
10|     R>
11|   THEN
12| ;
```

Listing 4: Choose sketch for Elementary Addition. Comments as in Listing 3 apply.

2. CHOOSE. Incorporating additional prior information, CHOOSE exactly specifies the results of the computation, namely the output of the first slot (line 5) is the carry, and the output of the second one (line 6) is the propagating result digit, both conditioned on the two digits and the carry on the data stack. It is described in Listing 4.

The rest of the sketch code reduces the problem size by one, and returns the solution by popping it from the return stack.

**Quantitative Evaluation on Addition** In set of experiments analogous to those in our evaluation on BubbleSort, we demonstrate the performance of $\partial 4$ on the addition task by examining test set sequence lengths of 8 and 64 while varying the lengths of the training set instances (Table 2). The Seq2Seq model again fails to generalize to longer sequences than those observed during training. In comparison, for the CHOOSE sketch $\partial 4$ learns the correct sketch behavior and generalizes to all test sequence lengths. However, for the less structured MANIPULATE sketch, the model is unable to converge when training on sequence lengths greater than one. We discuss this result, and similar difficulties which occurred in the sorting experiments, in the following section.

## 5 DISCUSSION

$\partial 4$ bridges the gap between a traditional programming language and a modern machine learning architecture. However, as we have seen in our evaluation experiments, faithfully simulating the underlying abstract machine architecture introduces its own unique set of concerns.

One such concern is the additional complexity of performing even simple tasks when they are viewed in terms of operations on the underlying machine state. As illustrated in Table 1, $\partial 4$ sketches can be effectively trained from the smallest of training sets, and generalize perfectly to sequences of any length. However, and perhaps unintuitively, difficulty arises when training from longer sequences of modest lengths. Even when dealing with relatively short training length sequences, the underlying machine can unroll into a problematically large number states. For problems whose machine execution is quadratic, like the sorting task (which at input sequences of length 4 has 120 machine states), we observe instabilities during training from backpropagating through such long RNN sequences.

While for many configurations we were able to train $\partial 4$ effectively on the sorting problem, we were unable to do so when using the MANIPULATE sketch on the addition problem. This is despite the addition experiments having comparatively shorter underlying execution RNNs, and highlights the

relationship between the computational difficulty of the problem and the degree of prior knowledge which must be provided in the sketch for successful learning. Using sketches with greater degrees of prior procedural knowledge $\partial 4$ is able to learn in difficult problem scenarios, but for sketches with high degrees of freedom we require an easier scenario for effective gradient propagation.

A second issue arises from incongruencies between how algorithmic data structures are typically used in a traditional language, and our newly introduced desire for them to learn behaviors which generalize to unseen data. For instance, it is common in Forth implementations of sequence processing (Sec. 3.3.1) to include the sequence length on the input, and to store it during computation for use in the algorithm. This is a sensible approach when working purely in the traditional programming paradigm, but in the context of learning it introduces information which influences the model's representations, prevents it from generalizing, and possibly leads to large memory requirements (when encoded with a one-hot vector). This motivates investigation into which traditional language properties are most suitable to this new hybrid paradigm, and which representations can be used to circumvent the problem.

## 6   RELATED WORK

**Program Synthesis**   The idea of program synthesis is as old as Artificial Intelligence, and has a long history in computer science (Manna & Waldinger, 1971). Whereas a large body of work has focused on using genetic programming (Koza, 1992) to induce programs from the given input-output specification (Nordin, 1997), there are also various Inductive Programming approaches (Kitzelmann, 2009) aimed at inducing programs from incomplete specifications of the code to be implemented (Albarghouthi et al., 2013; Solar-Lezama et al., 2006). We tackle the same problem of sketching, but in our case we fill the sketches with neural networks able to learn the slot behavior.

**Probabilistic and Bayesian Programming**   Our work is closely related to probabilistic programming languages such as Church (Goodman et al., 2008). They allow users to inject random choice primitives into programs as a way to define generative distributions over possible execution traces. In a sense, the random choice primitives in such languages correspond to the slots in our sketches. A core difference lies in the way we train the behaviour of slots: instead of calculating their posteriors using probabilistic inference, we estimate their parameters using backpropagation and gradient descent, similar to `TerpreT` (Gaunt et al., 2016), who induce code via backpropagation, and `Autograd` (Maclaurin et al., 2015), who enable automatic gradient computation in Python code. In addition, the underlying programming and probabilistic paradigm in these programming languages is often functional and declarative, whereas our approach focuses on a procedural and discriminative view. By using an end-to-end differentiable architecture, it is easy to seamlessly connect our sketches to further neural input and output modules, such as an LSTM that feeds into the machine heap, or a neural reinforcement learning agent that operates the neural machine. However, we leave connecting $\partial 4$ with neural upstream and downstream models for future work as it is out of the scope of this paper.

**Neural approaches**   Recently, there has been a surge of research in program synthesis, and execution in deep learning, with increasingly elaborate deep models. Many of these models were based on differentiable versions of abstract data structures (Joulin & Mikolov, 2015; Grefenstette et al., 2015; Kurach et al., 2015), and a few abstract machines, such as the NTM (Graves et al., 2014), Differentiable Neural Computers (Graves et al., 2016), and Neural GPUs (Kaiser & Sutskever, 2015). All these models are able to induce algorithmic behavior from training data. Our work differs in that our differentiable abstract machine allows us to seemingly integrate code and neural networks, and train the neural networks specified by slots via backpropagation through code interpretation.

The work in neural approximations to abstract structures and machines naturally leads to more elaborate machinery able to induce and call code or code-like behavior. Neelakantan et al. (2015a) learned SQL-like behavior—querying tables from natural language with simple arithmetic operations. Andreas et al. (2016) learn to *compose* neural modules to produce a desired behavior for a visual QA task. Neural Programmer-Interpreters (Reed & de Freitas, 2015) learn to represent and execute programs, operating on different modes of environment, and are able to incorporate decisions better captured in a neural network than in many lines of code (e.g. using image as an

input). Users inject prior procedural knowledge by training on program traces and hence require *full* procedural knowledge. In contrast, we enable users to use their partial knowledge in sketches.

Neural approaches to language compilation have also been researched, from compiling a language into neural networks (Siegelmann, 1994), over building neural compilers (Gruau et al., 1995) to adaptive compilation (Bunel et al., 2016). However, that line of research did not perceive neural interpreters and compilers as a means of injecting procedural knowledge as we did. To the best of our knowledge, $\partial 4$ is the first working neural implementation of an abstract machine for an actual programming language, and this enables us to inject such priors in a straightforward manner.

## 7 CONCLUSION AND FUTURE WORK

We have presented $\partial 4$, a differentiable abstract machine for the Forth programming language, and showed how it can be used to complement a programmer's prior knowledge through the learning of unspecified behavior in Forth sketches. The $\partial 4$ RNN successfully learns to sort and add, using only program sketches and program input-output pairs as input. We believe $\partial 4$, and the larger paradigm it helps establish, will be useful for addressing complex problems where low-level representations of the input are necessary, but higher-level reasoning is difficult to learn and potentially easier to specify.

In future work we aim to explore the relationship between backpropagation and programming language semantics. For instance, when performing a single conceptual operation whose implementation requires a series of simple operations on the machine state (e.g. Listing 4, line 7), one may wish to shortcut the flow of gradients around such low-entropy sequences using residual connections (Srivastava et al., 2015). An alternative solution to this problem would involve inducing hierarchies of actions, where a single more abstract action can be substituted for several repetitive actions, reducing the length of the unrolled action sequence. We also plan to apply $\partial 4$ to such problems in the NLP domain, like machine reading and knowledge base inference. In the long-term, we see the integration of non-differentiable transitions (such as those arising when interacting with a real environment), as an exciting future direction which sits at the intersection of reinforcement learning and probabilistic programming. Additionally, connecting $\partial 4$ with other differentiable models upstream and/or downstream is another direction we would like to tackle.

### ACKNOWLEDGMENTS

We thank Guillaume Bouchard, Dirk Weissenborn, Danny Tarlow, and the anonymous reviewers for fruitful discussions and helpful comments on previous drafts of this paper. This work was supported by a Microsoft Research PhD Scholarship, an Allen Distinguished Investigator Award, and a Marie Curie Career Integration Award.

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

APPENDIX

## A  FORTH WORDS

We implemented a small subset of available Forth words in $\partial 4$. The table of these words, together with their descriptions is given in Table 3. The commands are roughly divided into 6 groups. These groups, line-separated in the table, are:

**Data stack operations** {num}, 1+, 1-, DUP, SWAP, OVER, DROP

**Heap operations** @, !

**Comparators** >, <, =

**Return stack operations** >R, R>, @R

**Control statements** IF..ELSE..THEN, BEGIN..WHILE..REPEAT, DO..LOOP

**Subroutine control** :, {sub}, ;

Table 3: Forth words and their descriptions. TOS denotes top-of-stack, NOS denotes next-on-stack, DSTACK denotes the data stack, RSTACK denotes the return stack, and HEAP denotes the heap.

| Forth Word | Description |
|---|---|
| {num} | Pushes {num} to DSTACK. |
| 1+ | Increments DSTACK TOS by 1. |
| 1- | Decrements DSTACK TOS by 1. |
| DUP | Duplicates DSTACK TOS. |
| SWAP | Swaps TOS and NOS. |
| OVER | Copies NOS and pushes it on the TOS. |
| DROP | Pops the TOS (non-destructive). |
| @ | Fetches the HEAP value from the DSTACK TOS address. |
| ! | Stores DSTACK NOS to the DSTACK TOS address on the HEAP. |
| >, <, = | Consumes DSTACK NOS and TOS. Returns 1 (TRUE) if NOS $>$ $\vert$ $<$ $\vert$ = TOS respectivelly, 0 (FALSE) otherwise. |
| >R | Pushes DSTACK TOS to RSTACK TOS, removes it from DSTACK. |
| R> | Pushes RSTACK TOS to DSTACK TOS, removes it from RSTACK. |
| @R | Copies the RSTACK TOS TO DSTACK TOS. |
| IF..ELSE..THEN | Consumes DSTACK TOS, if it equals to a non-zero number (TRUE), executes commands between IF and ELSE. Otherwise executes commands between ELSE and THEN. |
| BEGIN..WHILE..REPEAT | Continually executes commands between WHILE and REPEAT while the code between BEGIN and WHILE evaluates to a non-zero number (TRUE). |
| DO..LOOP | Consumes NOS and TOS, assumes NOS as a limit, and TOS as a current index. Increases index by 1 until equal to NOS. At every increment, executes commands between DO and LOOP. |
| : | Denotes the subroutine, followed by a word defining it. |
| {sub} | Subroutine invocation, puts the program counter PC on RSTACK, sets PC to the subroutine address. |
| ; | Subroutine exit. Consumest TOS from the RSTACK and sets the PC to it. |

## B   BUBBLE SORT ALGORITHM DESCRIPTION

An example of a Forth program, that implements the Bubble sort algorithm, is shown in Listing 1. Here we provide a description of how the first iteration of this algorithm is executed by the Forth abstract machine:

The program begins at line 11, putting the sequence [2 4 2 7] on the data stack $D$, followed by the sequence length $4$[4]. It then calls the SORT word.

|   | $D$ | $R$ | $c$ | comment |
|---|---|---|---|---|
| 1 | [] | [] | 11 | execution start |
| 2 | [2 4 2 7 4] | [] | 8 | pushing sequence to $D$, calling SORT subroutine puts A$_{SORT}$ to $R$ |

For a sequence of length 4, SORT performs a do-loop in line 9 that calls the BUBBLE sub-routine 3 times. It does so by decrementing the top of $D$ with the 1− word to 3. Subsequently, 3 is duplicated on $D$ by using DUP, and 0 is pushed onto $D$.

|   | | | | |
|---|---|---|---|---|
| 3 | [2 4 2 7 3] | [A$_{SORT}$] | 9 | 1- |
| 4 | [2 4 2 7 3 3] | [A$_{SORT}$] | 9 | DUP |
| 6 | [2 4 2 7 3 3 0] | [A$_{SORT}$] | 9 | 0 |

DO consumes the top two stack elements 3 and 0 as the limit and starting point of the loop, leaving the stack $D$ to be [2,4,2,7,3]. We use the return stack $R$ as a temporary variable buffer and push 3 onto it using the word >R. This drops 3 from $D$, which we copy from $R$ with R@

|   | | | | |
|---|---|---|---|---|
| 7 | [2 4 2 7 3] | [Addr$_{SORT}$] | 9 | DO |
| 8 | [2 4 2 7] | [Addr$_{SORT}$ 3] | 9 | >R |
| 9 | [2 4 2 7 3] | [Addr$_{SORT}$ 3] | 9 | @R |

Next, we call BUBBLE to perform one iteration of the bubble pass, (calling BUBBLE 3 times internally), and consuming 3. Notice that this call puts the current program counter onto $R$, to be used for the program counter $c$ when exiting BUBBLE.

Inside the BUBBLE subroutine, DUP duplicates 3 on $R$. IF consumes the duplicated 3 and interprets is as TRUE. >R puts 3 on $R$.

|   | | | | |
|---|---|---|---|---|
| 10 | [2 4 2 7 3] | [A$_{SORT}$ 3 A$_{BUBBLE}$] | 0 | calling BUBBLE subroutine puts A$_{BUBBLE}$ to $R$ |
| 11 | [2 4 2 7 3 3] | [A$_{SORT}$ 3 A$_{BUBBLE}$] | 1 | DUP |
| 12 | [2 4 2 7 3] | [A$_{SORT}$ 3 A$_{BUBBLE}$] | 1 | IF |
| 13 | [2 4 2 7] | [A$_{SORT}$ 3 A$_{BUBBLE}$ 3] | 1 | >R |

Calling OVER twice duplicates the top two elements of the stack, to test them with <, which tests whether $2 < 7$. IF tests if the result is TRUE (0), which it is, so it executes SWAP.

|   | | | | |
|---|---|---|---|---|
| 14 | [2 4 2 7 2 7] | [A$_{SORT}$ 3 A$_{BUBBLE}$ 3] | 2 | OVER OVER |
| 15 | [2 4 2 7 1] | [A$_{SORT}$ 3 A$_{BUBBLE}$ 3] | 2 | < |
| 16 | [2 4 2 7] | [A$_{SORT}$ 3 A$_{BUBBLE}$ 3] | 2 | IF |
| 17 | [2 4 7 2] | [A$_{SORT}$ 3 A$_{BUBBLE}$ 3] | 2 | SWAP |

To prepare for the next call to BUBBLE we move 3 back from the return stack $R$ to the data stack $D$ via R>, SWAP it with the next element, put it back to $R$ with >R, decrease the TOS with 1− and invoke BUBBLE again. Notice that $R$ will accumulate the analysed part of the sequence, which will be recursively taken back.

|   | | | | |
|---|---|---|---|---|
| 18 | [2 4 7 2 3] | [A$_{SORT}$ 3 A$_{BUBBLE}$] | 3 | R > |
| 19 | [2 4 7 3 2] | [A$_{SORT}$ 3 A$_{BUBBLE}$] | 3 | SWAP |
| 20 | [2 4 7 3] | [A$_{SORT}$ 3 A$_{BUBBLE}$ 2] | 3 | >R |
| 21 | [2 4 7 2] | [A$_{SORT}$ 3 A$_{BUBBLE}$ 2] | 3 | 1- |
| 22 | [2 4 7 2] | [A$_{SORT}$ 3 A$_{BUBBLE}$ 2] | 0 | ...BUBBLE |

---

[4]Note that Forth uses Reverse Polish Notation and that the top of the data stack is 4 in this example.

When we reach the loop limit we drop the length of the sequence and exit SORT using the ; word, which takes the return address from $R$. At the final point, the stack should contain the ordered sequence [7 4 2 2].

## C  $\partial 4$ WORDS IMPLEMENTATION

Table 4: Forth words and their descriptions. TOS denotes top-of-stack, NOS denotes next-on-stack, DSTACK denotes data stack, and RSTACK denotes return stack.

| Symbol | Explanation |
|---|---|
| $\mathcal{M}$ | Stack, $\mathcal{M} \in \{\mathcal{D}, \mathcal{R}\}$ |
| $\mathbf{M}$ | Memory buffer, $\mathbf{M} \in \{\mathbf{D}, \mathbf{R}, \mathbf{H}\}$ |
| $\mathbf{p}$ | Pointer, $\mathbf{p} \in \{\mathbf{d}, \mathbf{r}, \mathbf{c}\}$ |
| $\mathbf{M}^{\star}$ | Increment and decrement matrices (circular shift). |
| | For $\star \in \{+, -\}$, $\mathbf{M}_{ij}^{\star} = \begin{cases} 1 & i \star 1 \equiv j (mod\ n)) \\ 0 & otherwise \end{cases}$ |
| **Pointer manipulation** | **Expression** |
| Increment $\mathbf{a}$ (or value $\mathbf{x}$) | $inc(\mathbf{a}) = \mathbf{a}^T \mathbf{M}^+$ |
| Decrement $\mathbf{a}$ (or value $\mathbf{x}$) | $dec(\mathbf{a}) = \mathbf{a}^T \mathbf{M}^-$ |
| Conditional jump $\mathbf{a}$ | $jump(\mathbf{c}, \mathbf{a}) : p = pop_{\mathbf{D}}() = TRUE \mathbf{c} \leftarrow p\mathbf{c} + (1-p)a$ |
| | $p = pop\ \mathbf{c} \leftarrow$ |
| $\mathbf{a}^{-1}$ | Next on stack, $\mathbf{a} \leftarrow \mathbf{a}^T \mathbf{M}^-$ |
| **Buffer manipulation** | |
| READ from $\mathbf{M}$ | $read_{\mathbf{M}}(\mathbf{a}) = \mathbf{a}^T \mathbf{M}$ |
| WRITE to $\mathbf{M}$ | $write_{\mathbf{M}}(\mathbf{x}, \mathbf{a}) : \mathbf{M} \leftarrow \mathbf{M} - \mathbf{a} \otimes \mathbf{1} \cdot \mathbf{M} + \mathbf{x} \otimes \mathbf{a}$ |
| PUSH $\mathbf{x}$ onto $\mathcal{M}$ | $push_{\mathbf{M}}(\mathbf{x}) : write_{\mathbf{M}}(\mathbf{x}, \mathbf{a})$ [side-effect: $\mathbf{d} \leftarrow inc(\mathbf{d})$] |
| POP an element from $\mathcal{M}$ | $pop_{\mathbf{M}}() = read_{\mathbf{M}}(\mathbf{a})$ [side-effect: $\mathbf{d} \leftarrow dec(\mathbf{d})$] |
| **Forth Word** | |
| Literal $\mathbf{x}$ | $push_{\mathbf{D}}(\mathbf{x})$ |
| `1+` | $write_{\mathbf{D}}(inc(read_{\mathbf{D}}(\mathbf{d})), \mathbf{d})$ |
| `1-` | $write_{\mathbf{D}}(dec(read_{\mathbf{D}}(\mathbf{d})), \mathbf{d})$ |
| `DUP` | $push_{\mathbf{D}}(read_{\mathbf{D}}(\mathbf{d}))$ |
| `SWAP` | $x = read_{\mathbf{D}}(\mathbf{d}), y = read_{\mathbf{D}}(\mathbf{d}^{-1})$ |
| | $: write_{\mathbf{D}}(\mathbf{d}, y) , write_{\mathbf{D}}(\mathbf{d}^{-1}, x)$ |
| `OVER` | $push_{\mathbf{D}}(read_{\mathbf{D}}(\mathbf{d}))$ |
| `DROP` | $pop_{\mathbf{D}}()$ |
| `@` | $read_{\mathbf{H}}(\mathbf{d})$ |
| `!` | $write_{\mathbf{H}}(\mathbf{d}, \mathbf{d}^{-1})$ |
| `<` | `SWAP >` |
| `>` | $e_1 = \sum_{i=0}^{n-1} i * \mathbf{d}_i, e_2 = \sum_{i=0}^{n-1} i * \mathbf{d}_i^{-1}$ |
| | $p = \phi_{pwl}(e_1 - e_2)$ *** (define piecewise linear f) |
| | $p\mathbf{1} + (p-1)\mathbf{0}$ |
| `=` | $p = \phi_{pwl}(\mathbf{d}, \mathbf{d}^{-1})$ |
| | $p\mathbf{1} + (p-1)\mathbf{0}$ |
| `>R` | $push_{\mathbf{R}}(\mathbf{d})$ |
| `R>` | $pop_{\mathbf{R}}()$ |
| `@R` | $write_{\mathbf{D}}(\mathbf{d}, read_{\mathbf{R}}(\mathbf{r}))$ |
| `IF..`$_1$`ELSE..`$_2$`THEN` | $p = pop_{\mathbf{D}}() = \mathbf{0}$ |
| | $p * ..\!_1 + (1-p) * ..\!_2$ |
| `BEGIN..`$_1$`WHILE..`$_2$`REPEAT` | $..\!_1\ jump(c, ..\!_2)$ |
| `DO..LOOP` | $: inc(p)\ p = p^{-1}\ jump(c, ..)\ jump(c, beginning)$ |

