# Peer review of "Programming With a Differentiable Forth Interpreter"

_ICLR 2017 — rejected_

[Official Review · AnonReviewer1 · rating 6 · confidence 4 · 16 Dec 2016]
**No Title**

This paper develops a differentiable interpreter for the Forth programming
language. This enables writing a program "sketch" (a program with parts left
out), with a hole to be filled in based upon learning from input-output
examples. The main technical development is to start with an abstract machine
for the Forth language, and then to make all of the operations differentiable.
The technique for making operations differentiable is analogous to what is done
in models like Neural Turing Machine and Stack RNN. Special syntax is developed
for specifying holes, which gives the pattern about what data should be read
when filling in the hole, which data should be written, and what the rough
structure of the model that fills the hole should be. Motivation for why one
should want to do this is that it enables composing program sketches with other
differentiable models like standard neural networks, but the experiments focus
on sorting and addition tasks with relatively small degrees of freedom for how
to fill in the holes.

Experimentally, result show that sorting and addition can be learned given
strong sketches.

The aim of this paper is very ambitious: convert a full programming language to
be differentiable, and I admire this ambition. The idea is provocative and I
think will inspire people in the ICLR community.

The main weakness is that the experiments are somewhat trivial and there are no
baselines. I believe that simply enumerating possible values to fill in the
holes would work better, and if that is possible, then it's not clear to me what
is practically gained from this formulation. (The authors argue that the point
is to compose differentiable Forth sketches with neural networks sitting below,
but if the holes can be filled by brute force, then could the underlying neural
network not be separately trained to maximize the probability assigned to any
filling of the hole that produces the correct input-output behavior?)

Related, one thing that is missing, in my opinion, is a more nuanced outlook of
where the authors believe this work is going. Based on the small scale of the
experiments and from reading other related papers in the area, I sense that it
is hard to scale up differentiable forth to large real-world problems. It
would be nice to have more discussion about this, and perhaps even an experiment
that demonstrates a failure case. Is there a problem that is somewhat more
complex than the ones that appear in the paper where the approach does not work?
What has been tried to make it work? What are the failure modes? What are the
challenges that the authors believe need to be overcome to make this work.

Overall, I think this paper deserves consideration for being provocative.
However, I'm hesitant to strongly recommend acceptance because the experiments
are weak.

[Official Review · AnonReviewer2 · rating 5 · confidence 2 · 19 Dec 2016]

This paper presents an approach to do (structured) program induction based on program sketches in Forth (a simple stack based language). They turn the overall too open problem of program induction into a slot filling problem, with a differentiable Forth interpreter, for which one can backprop through  the slots (as they are random variables). The point of having sketches/partial programs is that one can learn more complex programs than starting from scratch (with no prior information). The loss that they optimize (end to end through the program flow) is a L2 (RMSE) of the program memory (at targeted/non-masked adresses) and the desired output. They show that they can learn addition, and bubble sort, both with a Permute (3-way) sketch and with a Compare (2-way) sketch.

The idea of making a language fully differentiable to write partial programs (sketches) and have them completed was previously explored in the  probabilistic programming community and more recently with TerpreT. I think that using Forth (a very simple stack-based language) as the sketch definition language is interesting in itself, as it is between machine code (Neural Turing Machine, Stack RNN, Neural RAM approaches...) and higher level languages (Church, TerpreT, ProbLog...).

Section 3.3.1 (and Figure 2) could be made clearer (explain the color code, explain the parallel between D and the input list).

The experimental section is quite sparse, even for learning to sort, there is only one experimental setting (train on length 3 and test on length 8), and .e.g no study of the length at which the generalization breaks (it seems that it possibly does not), no study of the "relative runtime improvement" w.r.t. the training set (in size and length of input sequences). There are no baselines (not even at least exhaustive search, one of the neural approaches would be a plus) to compare to. Similarly, the "addition" experiment (section 4.2) is very shortly described, and there are no baselines either (whereas this is a staple of "neural approaches" to program induction). Does "The presented sketch, when trained on single-digit addition examples, successfully learns the addition, and generalises to longer sequences." mean that it generalizes to three digits or more?

Overall, the paper is very interesting, but it seems to me like the experiments do not support the claims, nor the usefulness, enough.

[Official Review · AnonReviewer3 · rating 7 · confidence 2 · 20 Dec 2016]
**No Title**

This paper presents an approach to make a programming language (Forth) interpreter differentiable such that it can learn the implementation of high-level instruction from provided examples. The paper is well-written and the research is well-motivated. Overall, I find this paper is interesting and pleasure to read.  However, the experiments only serve as proof of concept. A more detailed empirical studies can strength the paper. 

Comments:

- To my knowledge, the proposed approach is novel and nicely bridge programming by example and sketches by programmers. The proposed approach borrow some ideas from probabilistic programming and Neural Turing Machine, but it is significantly different from these methods. It also presents optimisations of the interpreter to speed-up the training. 

- It would be interesting to present results on different types of programming problems and see how complex of low-level code can be generated.

[Public Comment · Matko Bosnjak · 17 Jan 2017]
**New revision overview**

Dear reviewers, thank you for your thoughtful reviews. Based on your comments we uploaded an updated version of the paper in which we:
- added results of a Seq2Seq baseline to both sorting and adding tasks
- added a much more detailed quantitative analysis, regarding at which sequence length the models are not able to generalise, together with a qualitative evaluation
- expanded the addition task section with an additional sketch (choose), more text clarifying the sketches and a detailed quantitative analysis
- added a “Discussion” section clearly stating the limitations of the presented framework, and discussing failures
expanded the conclusion with additional future work, showing both nuanced outlook, and the long-term goals of the framework
- fixed Figure 2 and added details to section 3.3.1

[Final Decision · Program Chairs · 06 Feb 2017]
**ICLR committee final decision**

This work is stood out for many reviewers in terms of it's clarity ("pleasure to read") and originality, with reviewers calling it "very ambitious" and "provocative". Reviewers find the approach novel, and to fill an interesting niche in the area. All the reviewers were interested in the results, even if they did not buy completely the motivation (what "practically gained from this formulation", how does this fit in with prob programming).
 
 The main quality and impact issue is the lack of experimental results and baselines. Several reviewers find that the experiments "do not fit the claims", and ask for any type of baselines, even just enumeration. Lacking empirical evidence, there is a desire for a future plan showing what this type of approach could be useful for, even if it cannot really scale. I recommend this paper to be submitted to the workshop track.